# Randomised, controlled, feasibility trial comparing vasopressor infusion administered via peripheral cannula versus central venous catheter for critically ill adults: A study protocol

**Stacey Watts**[1], **Yogesh Apte**[1,2], **Thomas Holland**[1], **April Hatt**[1], **Alison Craswell**[1,3], **Frances Lin**[4,5], **Alexis Tabah**[2,6,7,8], **Robert Ware**[9], **Joshua Byrnes**[10], **Christopher Anstey**[9], **Gerben Keijzers**[9,11,12], **Mahesh Ramanan**[1,8,13,14]*

1 Caboolture Hospital, Caboolture, Australia, 2 James Mayne Academy of Critical Care, The University of Queensland, Brisbane, Australia, 3 School of Health, University of the Sunshine Coast, Sippy Downs, Australia, 4 Caring Futures Institute, College of Nursing and Health Sciences, Flinders University, Adelaide, Australia, 5 Sunshine Coast Health Institute, School of Health, University of the Sunshine Coast, Sippy Downs, Australia, 6 Intensive Care Unit, Redcliffe Hospital, Brisbane, Australia, 7 Faculty of Health, Queensland University of Technology, Brisbane, Australia, 8 Queensland Critical Care Research Network, Herston, Australia, 9 Menzies Health Institute Queensland and School of Medicine and Dentistry, Griffith University, Southport, Australia, 10 Centre for Applied Health Economics, School of Medicine and Dentistry, Griffith University, Southport, Australia, 11 Department of Emergency Medicine, Gold Coast Hospital and Health Service, Southport, Australia, 12 Faculty of Health Sciences and Medicine Bond University, Robina, Australia, 13 Critical Care Division, The George Institute for Global Health, University of New South Wales, Kensington, Australia, 14 Faculty of Health, Queensland of Technology, Brisbane, Australia

☯ SW and MR contributed equally to this work and should be considered joint first authors

* mahesh.ramanan@health.qld.gov.au

**Data Availability Statement:** No datasets were generated or analysed during the current study. All

## Abstract

### Background

When clinicians need to administer a vasopressor infusion, they are faced with the choice of administration via either peripheral intravenous catheter (PIVC) or central venous catheter (CVC). Vasopressor infusions have traditionally been administered via central venous catheters (CVC) rather than Peripheral Intra Venous Catheters (PIVC), primarily due to concerns of extravasation and resultant tissue injury. This practice is not guided by contemporary randomised controlled trial (RCT) evidence. Observational data suggests safety of vasopressor infusion via PIVC. To address this evidence gap, we have designed the "Vasopressors Infused via Peripheral or Central Access" (VIPCA) RCT.

### Methods

The VIPCA trial is a single-centre, feasibility, parallel-group RCT. Eligible critically ill patients requiring a vasopressor infusion will be identified by emergency department (ED) or intensive care unit (ICU) staff and randomised to receive vasopressor infusion via either PIVC or CVC. Primary outcome is feasibility, a composite of recruitment rate, proportion of eligible patients randomised, protocol fidelity, retention and missing data. Primary clinical outcome

relevant data from this study will be made available upon study completion.

**Funding:** • Emergency Medicine Foundation (Australasia) Queensland Program Grant ID: EMJS-411R37-2022-HOLLAND • The Prince Charles Hospital Foundation CKW-2022-02 • The University of Queensland: Mayne Academy of Critical Care Pilot Grant The funders had no role in study design, data collection and analysis, decision to publish, or preparation of the manuscript.

**Competing interests:** The authors have declared that no competing interests exist.

is days alive and out of hospital up to day-30. Secondary outcomes will include safety and other clinical outcomes, and process and cost measures. Specific aspects of safety related to vasopressor infusions such as extravasation, leakage, device failure, tissue injury and infection will be assessed.

## Discussion

VIPCA is a feasibility RCT whose outcomes will inform the feasibility and design of a multi-centre Phase-3 trial comparing routes of vasopressor delivery. The exploratory economic analysis will provide input data for the full health economic analysis which will accompany any future Phase-3 RCT.

## 1. Introduction

Vasopressor medications are used to restore haemodynamic stability and maintain blood pressure in patients with shock from various mechanisms [1]. Common indications in emergency medicine and critical care include sepsis and septic shock, trauma, cardiogenic shock, and to counteract vasodilatation from various drugs including sedative medications. Although early administration of vasopressors is significantly associated with increased (septic) shock control [2] they are not without adverse effects [3]. Central venous catheters (CVCs) are commonly inserted to facilitate administration of vasopressors however, of those patients who receive a CVC up to 19 percent develop some complications (including potentially serious ones such as infectious, mechanical and thrombotic complications) [4]. The urgency to commence vasopressors via a CVC poses logistical difficulties as safe placement of a CVC requires expertise, time and resources that may be difficult to mobilise expeditiously [5]. Particularly in an Emergency Department (ED) setting, where clinicians are faced with numerous competing interests, the time and resources required for CVC insertion may have negative consequences for the care of other patients requiring ED treatment.

The use of a peripheral intravenous catheter (PIVC) for administration of vasopressors is recommended in patients with a contraindication to a CVC [6]. The practice of commencing a vasopressor infusion via a PIVC was associated with improvements in processes of care, without increased risk of death and low extravasation rates and no events of tissue necrosis [7]. There is evidence that administration of vasopressors by PIVC has an acceptable safety profile with careful monitoring and safety precautions [7,8]. Despite an overall acceptable safety profile based on current evidence, administration of vasopressors via PIVC can potentially lead to complications [9] such as drug extravasation, skin and/or soft tissue necrosis, and inadequate drug delivery. The current evidence on tissue injury or extravasation from vasopressor administration via PIVCs is derived mainly from low-quality evidence [10]. A recent systematic review [11] reported that extravasation is uncommon and is unlikely to lead to major complications when vasopressors administered via PIVCs are given for a limited duration and under close observation.

There is substantial practice variation with regards to peripheral versus central delivery of vasopressors in ED/ICU practice, although traditionally the central route has been preferred. Institutional data [11] that there is a wide range of practice within our own institution but that most patients receive peripheral vasopressor infusions before going on to have a central line inserted. There were a smaller proportion of patients who received exclusively central or exclusively peripheral vasopressors. In the retrospective analysis, there were no differences in the

clinical outcomes such as mortality or length of stay, and the adverse events we found were generally minor (e.g., skin irritation, leakage from cannula). However, the patient baseline characteristics in the groups were different with many confounders and it is not possible to definitively establish the superiority of one technique over the other without a randomised trial. Therefore, there is substantial equipoise, both in terms of clinician practice, which is variable, and outcomes, which appear to be equivalent retrospectively.

Hence, when clinicians need to administer a vasopressor infusion, they are faced with the choice of administration via either PIVC or CVC, but there is a paucity of high-quality, contemporary randomised controlled trial (RCT) evidence to guide practice. Furthermore, with approximately 175,000 ICU admissions per annum across Australia, and 62,000 (35%) patients requiring vasopressors, this is a frequently encountered clinical dilemma [12]. To address this evidence gap, we have designed the "Vasopressors Infused via Peripheral or Central Access" (VIPCA) trial.

The primary objective of the VIPCA trial is to test whether, in critically ill patients with shock, it is feasible to conduct a definitive Phase-3 RCT comparing the efficacy and safety of administering vasopressors via PIVC versus CVC with a primary outcome of days alive and out of hospital at day-30 (DAH-30). A secondary outcome of the RCT will be to generate data to inform the design of a future RCT.

We hypothesise that conduct of a Phase-3 RCT will feasible as measured against pre-specified criteria.

## 2. Methods

The VIPCA feasibility trial was prospectively registered with the Australia New Zealand Clinical Trials Registry (Registration number: ACTRN12621000721808). This protocol (S1 Appendix) and statistical analysis plan are reported according to reporting guidelines for pilot trials from the 2019 SPIRIT statement [13]. On completion, trial results will be reported according to the CONSORT guidelines pilot extension (S6 Appendix) [14].

### 2.1 Design

The VIPCA trial is a single-centre, feasibility, parallel-group RCT. Eligible patients will be identified by emergency department (ED) or intensive care unit (ICU) staff and randomised to either the peripheral vasopressor group or central vasopressor group. It will be conducted at the ED and ICU in Caboolture Hospital, Metro North Hospital and Health Services, Queensland, Australia with recruitment having commenced in November 2022. At the anticipated recruitment rate of 1 patient per week, we had originally planned for completion of recruitment by September 2023. However, recruitment was slower than anticipated, and hence the trial is continuing recruitment at the time of writing.

### 2.2 Study population

All patients aged 18 years and over who present to the ED or ICU, who are deemed to require a vasopressor infusion by the treating clinician will be eligible for recruitment.

Patients will be excluded if they:

- are less than 18 years of age,

- are pregnant (confirmed or suspected),

- have received a vasopressor infusion for ≥4-hours,

- are requiring >0.1mcg/kg/min of Noradrenaline or equivalent at the time of screening,

- are requiring > 1 vasopressor agent,

- already have a CVC or peripherally inserted central catheter (PICC) in-situ,

- require CVC insertion for specific therapies other than vasopressors,

- are deemed to be ineligible for ICU admission or imminent death (i.e., within 24 hours, is strongly suspected by the treating clinician).

### 2.3 Randomisation

All patients being commenced on vasopressor infusions will be screened for VIPCA eligibility at the time of vasopressor commencement in both the ED and the ICU. Separate screening checklists and logs will be maintained in both locations. All clinical staff in both locations will be provided with ongoing trial education for the duration of recruitment and will be able to screen and enrol eligible patients.

Randomisation with allocation concealment will be performed using a pre-generated randomisation sequence (by the trial statistician) and sealed, opaque envelopes. Randomisation will be performed using randomised permuted blocks of size 2 and 4, and stratified by location of randomisation i.e., ED or ICU. Blinding is not feasible for this trial, as the presence of a CVC will be readily visible and known to staff, patients and families. Once randomised, patients will be treated according to their treatment allocation as soon as practically possible.

### 2.4 Ethics and governance

Ethical approval was obtained from The Prince Charles Hospital Health Human Research Ethics Committee (HREC/2021/QPCH/74377) for this trial to be conducted with a "consent to continue" model (S2 and S5 Appendices). Patients will be randomised when they meet eligibility and then consent will be obtained by the investigators or research staff at the earliest opportunity once the patient has regained capacity to provide consent. For patients who do not regain capacity to consent, informed consent will be sought from their official next of kin. The HREC approved a waiver of consent, where required, for the use of data from patients who are enrolled into the study but die before consent can be obtained.

An independent data safety monitoring committee (DSMC) was convened during the protocol development phase. The DSMC membership will include a senior clinician, senior researcher and a research coordinator with access to an independent statistician. They will monitor safety and adverse event data and perform a formal interim analysis after 20 patients have been recruited. A formal DSMC Charter will be developed to guide the DSMC. The DSMC will be independent of the trial sponsor and funders (S3 and S4 Appendices), and will be tasked with reviewing the safety data, particularly the serious adverse events arising from the trial. Early trial cessation for harm will be considered by the DSMC if there is sufficient evidence of increased trial-related complications due to extravasation of vasopressors.

### 2.5 Interventions

All included patients will receive standard medical care for their condition/s as determined by the treating clinician, including type and dose of vasopressors. The only aspect of patient care stipulated by this trial is route of vasopressor administration.

Patients will be randomised to either the peripheral vasopressor group or central vasopressor group. The peripheral vasopressor group will receive delivery of vasopressor infusion via PIVC, and delayed insertion of a CVC—that is, a CVC is not to be inserted for at least 12 hours from the time of randomisation. PIVCs used for vasopressor infusion will be of

minimum 20-gauge size, preferably 18-gauge, and inserted in the antecubital fossa or other large peripheral vein. For safety reasons, patients may receive a CVC earlier than 12 hours if one of the following conditions is met:

- noradrenaline equivalent dose requirement $\geq$0.2mcg/kg/min

- need for irritant medications/infusions that cannot be administered via a PIVC

- failure of drug delivery via PIVC

- complications of PIVC including extravasation, or tissue injury

The central vasopressor group will receive early insertion of a CVC for vasopressor infusion, that is, a CVC is to be inserted as soon as practical after randomisation. The target time to central delivery of vasopressor infusion is $\leq$4 hours from randomisation for the central vasopressor group.

All patients receiving vasopressor infusions will be closely monitored using standard department protocols in the ED and ICU (S7 and S8 Appendices).

### 2.6 Outcomes

**2.6.1 Feasibility outcomes.** The primary outcome for this trial is feasibility with pre-specified criteria. Feasibility will be individually determined by assessing the following:

- Recruitment rate $\geq$1 patient per week,

- $\geq$80% of eligible participants will be randomised

- Protocol fidelity $\geq$95% of participants in each of the allocated group will receive the intervention they were allocated within stipulated timeframes,

- Retention >95% of patients will consent to ongoing participation in the trial and <10% of patients will be lost to day-30 follow-up,

- Missing data: <10%.

**2.6.2 Clinical outcomes.** Being a feasibility trial, the range of outcomes assessed will be exploratory and used to inform the design of the Phase-3 trial, including the sample size calculation. As such, they will be reported with descriptive statistics only.

The clinical outcomes will be:

- Days alive and out of hospital up to day-30 post-randomisation (DAH-30)

- Complications related to CVC and PIVC (local, regional or systemic) during ED and ICU stay:

  - Need and reason for replacement

  - Extravasation of infused fluid into tissues

  - Leakage of infused fluid

  - Tissue injury including–

    - Skin erythema/irritation,

    - Skin necrosis,

    - Physician-determined need for phentolamine infiltration,

- Gangrene or other severe tissue injury requiring surgical intervention

- Central line associated blood stream infection

- Hospital and ICU length of stay

- Health related quality of life patient-reported outcome measure (PROM) using EuroQOL EQ-5D-5L at Day-30 follow-up

- Patient reported experience measure (PREM) using Australian Hospital Patient Experience Question Set (AHPEQS) [15] at Day-30 follow-up.

All follow-up and outcome assessment will be performed by trained research coordinators based at Caboolture Hospital. The trial processes are described in Fig 1.

**2.6.3 Process and cost measures.** A range of process and cost measures related to the trial interventions will also be evaluated:

- Number of peripheral venous puncture attempts

- Number of PIVCs inserted

- Number of CVCs inserted

- Time to commence vasopressor via CVC (in the CVC group)

- Healthcare costs including cost of device, staff time associated with insertion and monitoring, costs associated with subsequent complications and cost of hospital length of stay

## 2.7 Sample Size

Forty patients will be recruited (20 in each group), however no formal power calculations were performed, as this is feasibility trial and the superiority of one intervention over another is not being tested. This number has been deemed to be adequate for a feasibility trial [16].

## 2.8 Statistical analysis plan

The components of feasibility will be assessed using descriptive statistics against pre-specified benchmarks. There will be no pre-specified thresholds of statistical significance, nor will there be any formal sample size calculations. All analyses will be descriptive, with key feasibility and outcome data presented using appropriate graphs.

For all outcomes, descriptive statistics will be reported. Continuous Outcomes will be reported as either mean and standard deviation or median and inter-quartile range, depending in the distribution of the outcome variable. Categorical outcomes will be presented as frequency and percentage. The primary clinical outcome of DAH-30 will be compared between the groups using median regression and reported as median difference (95% confidence interval). Secondary outcomes measured using continuous data will be compared between-groups using either linear regression or median regression. While outcomes from binary variables will be compared using logistic regression. For all models the treatment group will be included as a main effect, and for repeated measures (e.g., health related quality of life utility score, EQ-5D) baseline values will be included as covariables. The within-group difference between the time points will be estimated when appropriate.

## 2.9 Economic analyses

Exploratory economic analyses will be performed using net monetary benefit of implementation as the outcome measures of interest. These will be performed primarily to inform the

| | STUDY PERIOD | | | | |
|---|---|---|---|---|---|
| | Enrolment | Allocation | Post-allocation | | Close-out |
| TIMEPOINT** | *Once vasopressor infusion has commenced* | *Once all inclusion/exclusion criteria satisfied* | *Daily during ICU[1] stay* | *Hospital discharge* | *Day-28 follow-up* |
| **ENROLMENT:** | | | | | |
| **Eligibility screen** | X | | | | |
| **Informed consent** | | X | | | |
| **Allocation** | | X | | | |
| **INTERVENTIONS:** | | | | | |
| *Early CVC[2] insertion* | | ●———————————● | | | |
| *Delayed CVC insertion* | | ●———————————● | | | |
| **ASSESSMENTS:** | | | | | |
| *Screening and baseline data* | | X | | | |
| *Complications related to vasopressor delivery* | | | X | X | |
| *Vasopressor use* | | | X | | |
| *ICU Treatments* | | | X | | |
| *Vascular access* | | | X | | |
| *Lengths of stay* | | | | X | |
| *Mortality* | | | X | X | X |
| *Health-related quality of life* | | | | | X |
| *Patient-reported outcome measures* | | | | | X |

1) ICU: Intensive Care Unit
2) CVC: Central Venous Catheter

**Fig 1. SPIRIT schedule.** 1) ICU: Intensive Care Unit 2) CVC: Central Venous Catheter.

design of a Phase-3 RCT. DAH-30 combined with health-related quality of life utility scores will be monetarised and included in the analysis using accepted threshold values for a quality adjusted life year.

A probabilistic decision model will be constructed to simulate the clinical pathways associated with the two groups. The preliminary model will identify all input parameters required for a full economic evaluation to be conducted alongside an adequately powered Phase-3 RCT and determine feasibility of data collection alongside the clinical trial, as well as additional sources and reliability of estimates of the required economic input parameters. The analysis will be from a health system perspective and consider the potential cost savings from differences in utilisation of devices and consumables (including staff time associated with procedures) as well as the subsequent cost of adverse events and complications.

Resource utilisation variables will be collected and supplemented with literature searches for other model values (for example cost of adverse events). Probabilistic sensitivity analysis will be used to characterise the uncertainty in the economic evaluation based on the results of the feasibility trial. Contribution to the overall uncertainty in the economic results from each model parameter will be explored using one-way sensitivity analyses.

## 2.10 Pre-specified nested study

A device selection and management sub-study will collect additional data for patients randomised to the peripheral vasopressor group during insertion and management of their peripheral intravenous site. Complications to PIVC, such as phlebitis and infiltration, interrupt treatment which can be distressing for patients and result in longer hospital stays [17]. Understanding decision making around PIVC insertion and management and the associated outcomes will assist in ensuring best practice for delivery of vasopressors via PIVC. When a PIVC is inserted, a member of either the study team, or another staff member trained in study procedures, will approach the operator who performed the procedure to complete a survey as soon as practicable after the insertion. A data collection tool was developed based on existing evidence [17,18].

**2.10.1 Sub-study outcomes.** The peripheral vasopressor group will include 20 participants. Therefore, the sub-study outcomes will be assessed and reported with descriptive statistics only. This data will assist to understand health professional decision making for PIVC insertion and management for delivery of peripheral vasopressors and inform the development of the larger trial.

Outcomes from this data will include:

- PIVC insertion attempt success

- PIVC device gauge/location/securement

- Insertion process characteristics

- Operator decision making

- PIVC outcomes (infection, use, removal)

## 2.11 Trial status

After obtaining all necessary approvals, the VIPCA Trial commenced on 16[th] November 2022. At the time of writing, 33 patients had been recruited. It is anticipated that recruitment will be completed by April 2024. A formal interim analysis was performed by the DSMC after recruitment of 20 patients to evaluate safety data. The DSMC advised continuation of the trial to completion as per protocol was advised.

## 3. Discussion

### 3.1 Key message

Vasopressor medications, including drugs such as noradrenaline, adrenaline and vasopressin, have traditionally been administered via CVCs, primarily due to concerns of extravasation and infiltration of vasopressors and resultant tissue injury. However, this practice is not currently guided by high-quality, contemporary RCT evidence. The VIPCA will begin the process of addressing this evidence gap by informing the feasibility and design of a multicentre Phase-3 trial comparing routes of vasopressor delivery. Our study shows that the VIPCA trial is feasible.

### 3.2 Significance

To enable design and conduct of such a Phase-3 trial, key feasibility data will be collected during the VIPCA trial. These data will enable us to design an appropriately powered trial which can address the central question of peripheral versus centrally administered vasopressors. Three of the feasibility outcomes are dedicated to recruitment and retention. Recruitment rate estimates are vital to planning, and will determine how many sites will be required for timely completion of a Phase-3 trial. For example, our target recruitment rate of $\geq 1$ patient per week has not been met so far during recruitment. Thus, we will use a more realistic recruitment rate estimate derived from our actual experience, along with the proportion of eligible patients recruited, during the VIPCA trial rather than an arbitrary estimate for designing any future trials in this field. Similarly, the retention rate and loss to follow-up rates will be used to inflate sample size for the Phase-3 appropriately. Protocol fidelity is important in determining whether patients are actually receiving their allocated intervention. Any issues with low protocol fidelity would have to be addressed with "practice-change" and research culture interventions before a Phase-3 trial could be attempted.

If a Phase-3 trial were to demonstrate that outcomes and serious adverse events were equivalent between the two groups, this would have major implications for clinical practice and the health system. There is potential for significant patient, economic and health service benefits if the use of CVCs can be safely reduced. For patients, avoidance of CVC insertion can have many benefits including avoiding a potentially painful, uncomfortable procedure and avoiding all the risks of central vein cannulation. There may also be reduction in delays to initiation of vasopressor infusions, potentially improving outcomes in conditions such as septic shock. Patients who have less severe shock and need lower doses and shorter durations of vasopressor infusion, and those who are unlikely to go on to require CVC for some other reason, stand to benefit the most. Clinicians can save time by using PIVCs, which can be inserted more quickly and by a wider range of healthcare professionals, including physicians of all grades and many nurses. The nested study outcomes will improve our understanding of healthcare professional decision making for PIVCs in the administration of high-risk vasopressor medication, an area of scant evidence. Insertion, site and device selection, securement and PIVC feasibility outcomes will add to the body of knowledge for clinicians already using the PIVC route of administration and to the development of future Phase-3 RCT. Comparatively, CVC insertion is usually performed by experienced critical care (ED/ICU/Anaesthesia) physicians. The healthcare system may benefit from reduced costs (CVC kits typically cost more than PIVCs) and increased efficiency. VIPCA will provide feasibility data, particularly on completeness of data collection of input parameters and baseline values for key variables (e.g., EQ-5D utility scores in this patient population), for a formal health economic analysis accompanying a future Phase-3 RCT. All these advantages may be of particular importance in low-resource settings,

where healthcare funding and staffing may be limited. We anticipate that results from a Phase-3 trial, if conducted, would be broadly applicable to a variety of healthcare settings, including in high- and low-middle income countries.

### 3.3 Strengths and limitations

VIPCA has been designed in accordance with best practice guidelines for the conduct of RCTs. It will provide key feasibility data which will inform the development of a definitive trial. It addresses a key clinical question of substantial importance for clinicians, patients and healthcare systems alike, and will produce broadly generalisable results for critically ill patients in EDs and ICUs globally.

VIPCA is limited by its small sample size, owing to its feasibility design, and thus will not by itself result in practice change. The nature of the intervention prevents blinding, though other aspects of methodological quality such as computer-generated stratified block randomisation sequence and allocation concealment have been incorporated. Bias mitigation strategies such as blinded outcome assessment were not possible due to funding constraints. However, we strictly selected objective outcome measures to reduce bias from lack of blinding.

## 4. Conclusion

VIPCA is a currently recruiting randomised controlled feasibility trial of PIVC versus CVC for administration of vasopressor infusion for critically ill patients with shock which will deliver key feasibility data to inform the design of a definitive Phase-3 RCT.

## Supporting information

**S1 Appendix. VIPCA Trial Protocol Version 3.0.**
(PDF)

**S2 Appendix. Ethics approval (The Prince Charles Hospital Human Research Ethics Committee).**
(PDF)

**S3 Appendix. Funding agreement (The Common Good Foundation).**
(PDF)

**S4 Appendix. Funding agreement (Emergency Medicine Foundation).**
(PDF)

**S5 Appendix. PLOS one human participants research checklist.**
(DOCX)

**S6 Appendix. SPIRIT checklist.**
(DOC)

**S7 Appendix. PIVC management.**
(PDF)

**S8 Appendix. CVC management.**
(PDF)

**S1 File.**
(DOCX)

**S2 File.**
(DOCX)

## Acknowledgments

MR acknowledges support from the Metro North Hospital and Health Services Clinician-Researcher Fellowship.

Sponsor
Metro North Office of Research
Metro North Hospital and Health Services
Level 7, Block 7
Royal Brisbane and Women's Hospital
Herston QLD Australia 4029

## Author Contributions

**Conceptualization:** Stacey Watts, Yogesh Apte, Christopher Anstey, Mahesh Ramanan.

**Funding acquisition:** Stacey Watts, Yogesh Apte, Thomas Holland, Mahesh Ramanan.

**Investigation:** April Hatt.

**Methodology:** Yogesh Apte, Thomas Holland, April Hatt, Alison Craswell, Alexis Tabah, Joshua Byrnes, Christopher Anstey, Gerben Keijzers, Mahesh Ramanan.

**Supervision:** Frances Lin, Robert Ware, Joshua Byrnes, Christopher Anstey, Gerben Keijzers.

**Writing – original draft:** Mahesh Ramanan.

**Writing – review & editing:** Stacey Watts, Yogesh Apte, Thomas Holland, April Hatt, Alison Craswell, Frances Lin, Alexis Tabah, Robert Ware, Joshua Byrnes, Christopher Anstey, Gerben Keijzers, Mahesh Ramanan.

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
