## [Decision Letter · Decision Letter 0]

7 Jan 2024

PONE-D-23-36879A randomised, controlled, feasibility trial comparing vasopressor infusion administered via peripheral cannula versus central venous catheter for critically ill adults: a study protocolPLOS ONE

Dear Dr. Ramanan,

Thank you for submitting your manuscript to PLOS ONE. After careful consideration, we feel that it has merit but does not fully meet PLOS ONE’s publication criteria as it currently stands. Therefore, we invite you to submit a revised version of the manuscript that addresses the points raised during the review process.

- Please correct identifying number

- Please explain in the manuscript why you exclude patients who “are requiring >0.1mcg/kg/min of Noradrenaline or equivalent at the time of screening”

- How many patients are included today (1 patient per week with start in November 2022 = 52 patients at that time…)

- Please give more details about how Norepinephrine infusion will be prepared? (electrical syringe? elastomeric pump?) How changeover will be performed?

We look forward to receiving your revised manuscript.

Kind regards,

Jean Baptiste Lascarrou

Academic Editor

PLOS ONE

Journal requirements: 1. When submitting your revision, we need you to address these additional requirements.Please ensure that your manuscript meets PLOS ONE's style requirements, including those for file naming. The PLOS ONE style templates can be found at https://journals.plos.org/plosone/s/file?id=wjVg/PLOSOne_formatting_sample_main_body.pdf and https://journals.plos.org/plosone/s/file?id=ba62/PLOSOne_formatting_sample_title_authors_affiliations.pdf. 2. Thank you for stating the following financial disclosure: 

 [•
Emergency Medicine Foundation (Australasia) Queensland Program Grant ID: EMJS-411R37-2022-HOLLAND 

•
The Prince Charles Hospital Foundation CKW-2022-02•
The University of Queensland: Mayne Academy of Critical Care Pilot Grant].  Please state what role the funders took in the study.  If the funders had no role, please state: ""The funders had no role in study design, data collection and analysis, decision to publish, or preparation of the manuscript."" If this statement is not correct you must amend it as needed. Please include this amended Role of Funder statement in your cover letter; we will change the online submission form on your behalf. 3. Please provide a complete Data Availability Statement in the submission form, ensuring you include all necessary access information or a reason for why you are unable to make your data freely accessible. If your research concerns only data provided within your submission, please write "All data are in the manuscript and/or supporting information files" as your Data Availability Statement. 4. Please include the reference section of your manuscript. 5. Your ethics statement should only appear in the Methods section of your manuscript. If your ethics statement is written in any section besides the Methods, please move it to the Methods section and delete it from any other section. Please ensure that your ethics statement is included in your manuscript, as the ethics statement entered into the online submission form will not be published alongside your manuscript. 

Reviewers' comments:

Reviewer's Responses to Questions

**Comments to the Author**

1. Does the manuscript provide a valid rationale for the proposed study, with clearly identified and justified research questions?

Reviewer #1: Yes

Reviewer #2: Yes

Reviewer #3: Yes

2. Is the protocol technically sound and planned in a manner that will lead to a meaningful outcome and allow testing the stated hypotheses?

Reviewer #1: Yes

Reviewer #2: Partly

Reviewer #3: Yes

3. Is the methodology feasible and described in sufficient detail to allow the work to be replicable?

Reviewer #1: Yes

Reviewer #2: No

Reviewer #3: Yes

4. Have the authors described where all data underlying the findings will be made available when the study is complete?

Reviewer #1: No

Reviewer #2: No

Reviewer #3: Yes

5. Is the manuscript presented in an intelligible fashion and written in standard English?

Reviewer #1: Yes

Reviewer #2: No

Reviewer #3: Yes

6. Review Comments to the Author

You may also provide optional suggestions and comments to authors that they might find helpful in planning their study.

Reviewer #1: As far as I'm concerned, there's no major problem with the way this article has been written.

Except for two points:

- 1 It seems to me very important to add to the section "2.5 Interventions", as stated in the protocol, the criteria for waiving the delayed insertion of a CVC. Without these details, patient safety conditions do not seem to be met within the framework of this protocol.

"iii. A CVC can be inserted earlier than 12 hours if required for the following reasons:

▪ Noradrenaline-equivalent dose ≥0.2mcg/kg/min,

▪ Need for irritant medications/infusions that cannot be administered via a PIVC,

▪ Failure of drug delivery via PIVC,

▪ Complications of PIVC including extravasation of VPI, or tissue necrosis."

- 2 There is a spelling error in the "2.5 Interventions" section:

20-guage size, preferably 18-guage, it's gauge and not gauge

And finally, the conclusion seems a bit rushed to me, even though it perfectly sums up the idea of this research project.

Reviewer #2: Thanks for inviting me to review this reasibility RCT protocol on peripheral vs central vasopressors at a single institution. The manuscript is reasonably well written, however there are stylistic and language errors throughout, and even a few spelling mistakes, and I encourage the authors have a closer read of the text.

My main concern is the primary clinical outcome, which is DAOH-30. Given that both arms will be receiving vasopressors, I'm not sure that there will be a meaningful difference in this outcome, even if this study is extended to a Phase 3 trial. Both arms will have pressors titrated to effect, so would it not make sense for the primary clinical outcome to be something that will generate more clinical difference? eg safety-related complications

Furthermore, there is insufficient details regarding the safety monitoring of PIVCs, which is fundamentally the key concern with PIVC vasopressors. This needs to be protocolised, and has not been well detailed in this manuscript. Details re complication monitoring/definitions are also sparse.

Some limited minor issues:

Abstract

- RC is likely to mean to be RCT

Introduction

- citation 4 is described as a systematic reivew yet this appears to be an editorial??

- Citation 7 is also described as a SR yet this is a single institutional review (as per the title)

- 'Although administration of vasopressor infusion via a PIVC is not associated with increased morbidity it can lead to complications' - I would argue that this is contradictory - I understand what the authors presumably mean (ie infusion of vasopressors through a peripheral vein via patent non-extravasated PIVC is not associated with increased morbidity), but I would contend that administration of PIVC vasopressor is a holistic process, and the risk of extravasation should be counted as morbidity associated with vasopressor infusion. I would suggest rephrasing this sentence

Methods

- the ANZCTR registration number is wrong.

- statistical plans re early terminination by the DSMC should be detailed

- details re PIVC monitoring need to be included and standardized. What about USS insertion of PIVC? Long PIVCs? Is there a need to confirm PIVC patency via flushing etc? These are important protocol details and should be outlined. Also gauge is spelt wrong.

Reviewer #3: Your article provides a comprehensive overview of the VIPCA feasibility trial, addressing the pertinent issues surrounding vasopressor administration through peripheral intravenous catheters (PIVCs) versus central venous catheters (CVCs). The protocol is meticulously detailed, outlining the trial's design, methods, and objectives. The language is predominantly formal, which suits the scientific context but could benefit from occasional simplification for broader accessibility.

The study's significance is well articulated, emphasizing the potential impact on clinical practices and healthcare systems. The strengths and limitations are candidly discussed, with acknowledgment of the trial's small sample size limitation. Overall, the document serves its purpose of detailing the VIPCA trial while providing room for some language refinement for enhanced clarity and accessibility.

The article's strengths lie in its emphasis on the need for a randomized controlled trial (RCT) and the acknowledgment of confounding factors in retrospective analysis.

The comprehensive patient exclusion criteria are highlighted, but further clarification is needed for certain aspects. Feasibility outcomes are briefly discussed, yet a more elaborate rationale for each criterion would enhance understanding. The challenge of blinding is acknowledged, and suggestions for mitigating biases could be explored. The economic analyses section, while mentioned briefly, needs expansion to address the trial's potential impact on healthcare costs and its role in shaping the Phase-3 RCT design. Overall, refining these aspects would strengthen the article's clarity and provide a more robust foundation for the VIPCA trial.

Specific Comments from Reviewer to Authors:

Page 7 Line 139:

- Consider specifying the publication date or version of the SPIRIT statement

Page 7 141-142:

- Ensure consistency in spelling: "Metro North Hospital" or "Metro North Hospital and Health Services Office."

Page 7 147-151:

- Specify the anticipated duration of the trial.

Page 7 150:

- Clarify if Caboolture Hospital is part of Metro North Hospital and Health Services.

Page 10 204-206:

- Specify the reason for the minimum 20-gauge size for PIVCs, and why 18-gauge is preferable.

Page 10 211-221:

- Clarify whether the feasibility outcomes will be measured independently or in combination.

Page 10-11 222 – 246:

- For clinical outcomes, consider providing information on how the outcomes will be measured (e.g., instruments used for PROMs/PREMs).

Page 14 325-329:

- Specify the estimated duration of the trial.

Page 14 331-337:

- Specify the types of vasopressors considered for clarity.

Page 14-15 Lines 338-362:

- To enhance clarity, specifying the patient population that stands to benefit most from reduced CVC use would be valuable.

7. PLOS authors have the option to publish the peer review history of their article (what does this mean?). If published, this will include your full peer review and any attached files.

Reviewer #1: **Yes: **Julien Le Roy

Reviewer #2: No

Reviewer #3: **Yes: **Joseph Varon MD

---

## [Author Response · Author response to Decision Letter 0]

7 Feb 2024

Dear Professor Jean Baptiste Lascarrou,

Thank you for reviewing our manuscript and providing an opportunity to submit a revision.

We have addressed every comment from the Editorial Office and the Reviewers. Our responses can be seen in yellow highlight below.

We hope that you will now find the manuscript suitable for publication. We are more than happy to provide any further information requested.

Sincerely,

Mahesh Ramanan and Stacey Watts on behalf of all authors

01 Feb 2024

 

Editor Comments

- Please correct identifying number

Response: Yes. This has been done. 

- Please explain in the manuscript why you exclude patients who “are requiring >0.1mcg/kg/min of Noradrenaline or equivalent at the time of screening”

Response: This is a practical exclusion. Clinicians at the participating institution are likely to add a second vasoactive agent (such as adrenaline or vasopressin) when the noradrenaline dose is in the 0.1-0.15mcg/kg range. They are also likely to insert a CVC once the noradrenaline dose escalates beyond the 0.1mcg/kg threshold. Therefore, from a practical viewpoint, enrolling these patients into the trial is likely to be unacceptable to clinicians and protocol compliance is likely to be low, particularly among patients randomised to delayed CVC insertion.

- How many patients are included today (1 patient per week with start in November 2022 = 52 patients at that time…)

Response: This information has now been added to the manuscript.

- Please give more details about how Norepinephrine infusion will be prepared? (electrical syringe? elastomeric pump?) How changeover will be performed?

Response: All noradrenaline infusions are delivered via BD Alaris™ Infusion System electronic infusion pumps. Changeover from PIVC to CVC are performed with a two-minute (or shorter, if hypertension develops) overlap, i.e., both PIVC and CVC infusions are delivered for a two-minute period following which the PIVC infusion is ceased. 

Please ensure that your manuscript meets PLOS ONE's style requirements, including those for file naming. The PLOS ONE style templates can be found at https://journals.plos.org/plosone/s/file?id=wjVg/PLOSOne_formatting_sample_main_body.pdf and https://journals.plos.org/plosone/s/file?id=ba62/PLOSOne_formatting_sample_title_authors_affiliations.pdf.

Response: Yes we have now done this. 

 [• Emergency Medicine Foundation (Australasia) Queensland Program Grant ID: EMJS-411R37-2022-HOLLAND 

• The Prince Charles Hospital Foundation CKW-2022-02

• The University of Queensland: Mayne Academy of Critical Care Pilot Grant]. 

Please state what role the funders took in the study. If the funders had no role, please state: ""The funders had no role in study design, data collection and analysis, decision to publish, or preparation of the manuscript."" If this statement is not correct you must amend it as needed. 

Response: Thank you. We have added this statement.

Response: We have completed this information in the Data Availability Statement. 

4. Please include the reference section of your manuscript.

Response: The Reference section is included at the end of manuscript. 

Response: The ethics information is only contained in the Methods section.

Response: This has now been added.

Reviewer #1: 

As far as I'm concerned, there's no major problem with the way this article has been written.

Response: Thank you for reviewing our manuscript and helpful comments.

Except for two points:

- 1 It seems to me very important to add to the section "2.5 Interventions", as stated in the protocol, the criteria for waiving the delayed insertion of a CVC. Without these details, patient safety conditions do not seem to be met within the framework of this protocol.

"iii. A CVC can be inserted earlier than 12 hours if required for the following reasons:

▪ Noradrenaline-equivalent dose ≥0.2mcg/kg/min,

▪ Need for irritant medications/infusions that cannot be administered via a PIVC,

▪ Failure of drug delivery via PIVC,

▪ Complications of PIVC including extravasation of VPI, or tissue necrosis."

Response: Yes thanks for pointing out. It is written in the protocol, but was omitted from the manuscript. Now rectified.

- 2 There is a spelling error in the "2.5 Interventions" section:

20-guage size, preferably 18-guage, it's gauge and not gauge

Response: Thank you. This has been corrected.

And finally, the conclusion seems a bit rushed to me, even though it perfectly sums up the idea of this research project.

Reviewer #2: 

Thanks for inviting me to review this reasibility RCT protocol on peripheral vs central vasopressors at a single institution. The manuscript is reasonably well written, however there are stylistic and language errors throughout, and even a few spelling mistakes, and I encourage the authors have a closer read of the text.

Response: Thank you for your detailed review and suggestions.

My main concern is the primary clinical outcome, which is DAOH-30. Given that both arms will be receiving vasopressors, I'm not sure that there will be a meaningful difference in this outcome, even if this study is extended to a Phase 3 trial. Both arms will have pressors titrated to effect, so would it not make sense for the primary clinical outcome to be something that will generate more clinical difference? eg safety-related complications

Response: Thank you. Yes, is something that is currently under consideration for potential Phase 3 work. As far as this current trial is concerned, the primary outcome is feasibility. We selected the primary clinical outcome of DAH-30 to gather feasibility data. We anticipate that the DAH-30 will be similar between groups, as you have explained. It is possible that if we go ahead with a Phase 3 RCT with DAH-30 as the primary outcome, it would have to be a non-inferiority trial. This in itself would be very meaningful for patients and clinicians, because it would help us move away from the dogma that CVCs must be inserted for vasopressor infusions. The other option is to have a composite safety outcome. The reported incidence of patient-important adverse effects from using PIVC for vasopressors is quite low. However, most of this data is retrospective. The prospectively collected data from VIPCA will give us a better idea of whether safety could be a primary superiority outcome. 

Furthermore, there is insufficient details regarding the safety monitoring of PIVCs, which is fundamentally the key concern with PIVC vasopressors. This needs to be protocolised, and has not been well detailed in this manuscript. Details re complication monitoring/definitions are also sparse.

Response: Yes this is a very important part of the trial. We have standard protocols for monitoring of patients with vasopressor infusion via PIVC, and a general protocol for CVC. These are referenced in the Study Protocol, but were not included with the original submission. We have now added them as Supporting Information and a sentence has been added to section 2.5 in Methods.

Some limited minor issues:

Abstract

- RC is likely to mean to be RCT

Response: Fixed.

Introduction

- citation 4 is described as a systematic reivew yet this appears to be an editorial??

- Citation 7 is also described as a SR yet this is a single institutional review (as per the title)

Response: Apologies, some errors in referencing crept in from older versions of the manuscript. These have now been corrected.

- 'Although administration of vasopressor infusion via a PIVC is not associated with increased morbidity it can lead to complications' - I would argue that this is contradictory - I understand what the authors presumably mean (ie infusion of vasopressors through a peripheral vein via patent non-extravasated PIVC is not associated with increased morbidity), but I would contend that administration of PIVC vasopressor is a holistic process, and the risk of extravasation should be counted as morbidity associated with vasopressor infusion. I would suggest rephrasing this sentence

Response: Thanks. We have clarified this by stating that while there is an overall acceptable safety profile based on available data, complications can occur…..

Methods

- the ANZCTR registration number is wrong.

Response: The last 8 was missing in the methods, but was correct in the abstract. This has now been fixed.

- statistical plans re early terminination by the DSMC should be detailed

Response: A DSMC Charter was formally developed to guide the DSMC deliberations. We have expanded the section on DSMC in the methods. 

- details re PIVC monitoring need to be included and standardized. What about USS insertion of PIVC? Long PIVCs? Is there a need to confirm PIVC patency via flushing etc? These are important protocol details and should be outlined. Also gauge is spelt wrong.

Response: These are all important considerations that were addressed in the Appendices of the Study Protocol, which were not included with the original submission. They have now been included in the supporting information. In short, yes, we have specific guidelines on PIVC insertion and maintenance, as well as on monitoring.

Reviewer #3: Your article provides a comprehensive overview of the VIPCA feasibility trial, addressing the pertinent issues surrounding vasopressor administration through peripheral intravenous catheters (PIVCs) versus central venous catheters (CVCs). The protocol is meticulously detailed, outlining the trial's design, methods, and objectives. The language is predominantly formal, which suits the scientific context but could benefit from occasional simplification for broader accessibility.

The study's significance is well articulated, emphasizing the potential impact on clinical practices and healthcare systems. The strengths and limitations are candidly discussed, with acknowledgment of the trial's small sample size limitation. Overall, the document serves its purpose of detailing the VIPCA trial while providing room for some language refinement for enhanced clarity and accessibility.

The article's strengths lie in its emphasis on the need for a randomized controlled trial (RCT) and the acknowledgment of confounding factors in retrospective analysis.

The comprehensive patient exclusion criteria are highlighted, but further clarification is needed for certain aspects. Feasibility outcomes are briefly discussed, yet a more elaborate rationale for each criterion would enhance understanding. 

Response: The rationale for feasibility outcomes has now been added to the Discussion.

The challenge of blinding is acknowledged, and suggestions for mitigating biases could be explored. 

Response: Where feasible, blinded outcome assessment may be useful in mitigating some of this bias. We did not have the funding to enable us to achieve this, as research staff involved in screening, randomisation and consent were also involved in outcome assessment. We did choose objective outcome measures which may reduce some of the bias introduced by lack of blinding. These points have been added to the strengths and limitations section.

The economic analyses section, while mentioned briefly, needs expansion to address the trial's potential impact on healthcare costs and its role in shaping the Phase-3 RCT design.

Response: The economic analysis we have proposed is very much exploratory. It has been designed to inform planning for a full health economic analysis (indeed if this is feasible) alongside a future Phase-3 RCT. The main objective is to determine the completeness of collection of input parameters, and estimates of key variables in this study population (e.g., EQ-5D utility scores). 

Overall, refining these aspects would strengthen the article's clarity and provide a more robust foundation for the VIPCA trial.

Response: Thank you for you detailed review and helpful comments.

Specific Comments from Reviewer to Authors:

Page 7 Line 139:

- Consider specifying the publication date or version of the SPIRIT statement

Response: This has been added.

Page 7 141-142:

- Ensure consistency in spelling: "Metro North Hospital" or "Metro North Hospital and Health Services Office."

Response: This has been clarified. Please note that the research office is officially called the “Metro North Research Office” (does not have the “Hospital and Health Services”).

Page 7 147-151:

- Specify the anticipated duration of the trial.

Response: This has been added.

Page 7 150:

- Clarify if Caboolture Hospital is part of Metro North Hospital and Health Services.

Response: This has been added.

Page 10 204-206:

- Specify the reason for the minimum 20-gauge size for PIVCs, and why 18-gauge is preferable.

Response: This is purely due to PIVC length and concerns about “tissue-ing”. The PIVCs we stock (BD Insyte™ Autoguard™)reduce in length below 20-guage-: 22-guage has 25mm length while 20- and 18-guage have 30mm length with 48mm also stocked). Shorter length PIVC may intuitively have increased the risk of “tissued” PIVC, which we are keen to avoid.

Page 10 211-221:

- Clarify whether the feasibility outcomes will be measured independently or in combination.

Response: This has been added.

Page 10-11 222 – 246:

- For clinical outcomes, consider providing information on how the outcomes will be measured (e.g., instruments used for PROMs/PREMs).

Response: The instruments have been listed. For PROM, we are using the EQ-5D—5L at 30 days follow-up. For PREM, we are using the AHPEQS tool. Both have accompanying references.

Page 14 325-329:

- Specify the estimated duration of the trial.

Response: This has been added.

Page 14 331-337:

- Specify the types of vasopressors considered for clarity.

Response: This has been added.

Page 14-15 Lines 338-362:

- To enhance clarity, specifying the patient population that stands to benefit most from reduced CVC use would be valuable.

Response: This has been added.

---

## [Decision Letter · Decision Letter 1]

9 Apr 2024

A randomised, controlled, feasibility trial comparing vasopressor infusion administered via peripheral cannula versus central venous catheter for critically ill adults: a study protocol

PONE-D-23-36879R1

Dear Dr. Ramanan,

We’re pleased to inform you that your manuscript has been judged scientifically suitable for publication and will be formally accepted for publication once it meets all outstanding technical requirements.

Kind regards,

Jean Baptiste Lascarrou

Academic Editor

PLOS ONE

Additional Editor Comments (optional):

Reviewers' comments:

Reviewer's Responses to Questions

**Comments to the Author**

1. Does the manuscript provide a valid rationale for the proposed study, with clearly identified and justified research questions?

Reviewer #3: Yes

2. Is the protocol technically sound and planned in a manner that will lead to a meaningful outcome and allow testing the stated hypotheses?

Reviewer #3: Yes

3. Is the methodology feasible and described in sufficient detail to allow the work to be replicable?

Reviewer #3: Yes

4. Have the authors described where all data underlying the findings will be made available when the study is complete?

Reviewer #3: Yes

5. Is the manuscript presented in an intelligible fashion and written in standard English?

Reviewer #3: Yes

6. Review Comments to the Author

You may also provide optional suggestions and comments to authors that they might find helpful in planning their study.

Reviewer #3: Thank you for addressing my queries.

7. PLOS authors have the option to publish the peer review history of their article (what does this mean?). If published, this will include your full peer review and any attached files.

Reviewer #3: **Yes: **Joseph Varon, MD, FACP, FCCP, FCCM, FRSM

---

## [Editor Report · Acceptance letter]

30 Apr 2024

PONE-D-23-36879R1 

PLOS ONE

Dear Dr. Ramanan, 

I'm pleased to inform you that your manuscript has been deemed suitable for publication in PLOS ONE. Congratulations! Your manuscript is now being handed over to our production team.

Kind regards, 

on behalf of

Dr. Jean Baptiste Lascarrou 

Academic Editor

PLOS ONE